# An Empirical Analysis of Cashless Payment Systems for Business Transactions

**Mahfuzur Rahman** [1,*] , **Izlin Ismail** [2] , **Shamshul Bahri** [3] **and Muhammad Khalilur Rahman** [4]

1    Department of Finance and Economics, College of Business Administration, University of Sharjah,
     Sharjah 27272, United Arab Emirates
2    Department of Finance, Faculty of Business and Economics, Universiti Malaya,
     Kuala Lumpur 50603, Malaysia
3    Department of Decision Sciences, Faculty of Business and Economics, Universiti Malaya,
     Kuala Lumpur 50603, Malaysia
4    Faculty of Entrepreneurship and Business, Angkasa-Umk Research Academy, Universiti Malaysia Kelantan,
     Pengkalan Chepa 16150, Malaysia
*    Correspondence: mrahman@sharjah.ac.ae

**Abstract:** This paper examines the antecedents of cashless payment systems among businesses in Malaysia. The adoption of cashless payment systems by businesses has the potential to reduce the costs related to handling huge amounts of cash in the market and enhance transaction speed. Unfortunately, its current adoption in Malaysia is still small and very little is known about the factors. A seven-factor model based on the TOE framework was developed and tested. The partial least square (PLS) statistical approach was employed to analyze data collected from 200 business entities in Malaysia. The results reveal that compatibility and technology competence have higher significant relationships with the adoption of cashless payment systems. Management support, firm critical mass, competitive pressure, and information intensity are significantly related to the adoption of cashless payment systems, while firm size is not associated with it. The findings of this study provide significant practical implications for Malaysian stakeholders and technology vendors to recognize factors that affect a firm's adoption of cashless payment systems to support business transactions. By investigating the phenomenon through the TOE framework, this study presents an integrated model of cashless payment systems by businesses. Our findings also offer guidance for future application of the PLS method in cashless payment and related research. The paper provides a more holistic understanding of the factors influencing cashless payment systems among businesses.

**Keywords:** cashless payment adoption; partial least square; TOE framework; businesses

## 1. Introduction

Cashless payment systems present enormous benefits to businesses and the economy. For businesses, the ease of transactions via various modes of payment can increase revenue, improve operational efficiency, and lower operating costs [1,2]. Cashless payments are also seen as more hygienic for food vendors [3]. Cashless modes for small payments, such as Near Field Communication (NFC) technology, are found to be able to reduce queueing and the need to carry cash for high-volume and low-value transactions [4,5]. In a recent study, Kilay, Simamora and Putra [6] show that the supply chain performance of micro, small, and medium enterprises in Indonesia is directly correlated with the use of e-payment services.

Mohamad and Kassim's [7] study postulated that the adoption of e-payment services by micro-entrepreneurs (which make up more than 70% of the SME sector) could enhance their financial inclusion, as they tend to be seen as an unprofitable community. Despite these benefits, businesses' take-up of cashless payment systems has been slow. A study by Srouji [8] shows that cash and cheques still dominate payment settlements. In addition, they found that the firms' perception of banks' handling of electronic payment transactions

was less satisfactory. Meanwhile, another study found that the use of debit cards accounted for only 4 per cent of retail transactions, while credit cards comprised nearly 22 per cent [9]. As only 29 per cent of Malaysians own credit cards [10] there is still a huge potential for the use of cashless payment systems. One of the ways to increase the adoption of cashless payment systems within the retail industry is through point-of-sale (POS) terminals. Their employment within small businesses can transform the retail experience in the country. Unfortunately, very little is known about the factors that can influence the adoption of cashless payment systems among businesses in Malaysia. Meanwhile, studies in other countries suggest that firm size is an important antecedent of the technology's adoption [11,12]. They also suggest that innovation in payment solutions and the quality of the infrastructure are other antecedents [13,14]. Studies also suggested that businesses should adopt an open innovation viewpoint to accelerate their business performance [6,15].

Hence, this study aims to investigate the factors that influence the adoption of cashless payments by businesses in Malaysia, specifically the smaller businesses in urban areas that can access the latest payment technologies. These research questions are offered: (a) what are the factors that influence businesses' adoption of cashless payment systems in moving towards a cashless Malaysia? and (b) which factors are either strongly or weakly linked to the adoption of cashless payment systems?

To answer the above research questions, we have employed the TOE framework [16,17]. It is a suitable framework as it has been employed to investigate the adoption of e-procurement systems [18], electronic supply chain management systems [19], and enterprise resource planning systems in the communication industry [20]. The framework consists of three dimensions: technological, organizational, and environmental [21]. Furthermore, the framework has not been employed previously to investigate businesses' adoption of cashless payment systems. Using the model, these factors were revealed as relevant for this study: compatibility under the technological factor; firm size, management support, and technological competence under the organizational factor; and information intensity under the environmental factor. This model, hopefully, will inform businesses about the technological, organizational, and environmental challenges facing cashless payment systems.

The rest of the paper is organized as follows: Section 2 reviews the extensive literature and proposes hypotheses for this research; Section 3 describes the research methodology; Section 4 illustrates the results and data analysis; Section 5 discusses the empirical findings and their implications; and finally, Section 6 concludes and discusses limitations and suggestions for future research.

## 2. Literature Review

### 2.1. Compatibility

Compatibility is referred to as the degree to which a particular technological innovation fits the current values, previous experience, and issues considered essential by potential adopters [17]. Businesses may be deterred from adopting cashless payments due to the incompatibility with their business operations and the different cashless payment systems technology's lack of standardization. Compatibility is a construct within the technological factor, which entails values, experience, norms, and practices. According to Upward and Jones [22], compatibility consists of operational and normative compatibility. Technical compatibility can treat work procedures and values for business compatibility. Wong et al. [21] identified compatibility as a technological factor, and it is derived from the innovation diffusion theory. In the adoption of a cashless society, compatibility is the consistency between innovation and its values [23,24]. Previous studies [25,26] have indicated that compatibility is a crucial predictor for the adoption of internet payment systems. According to Świecka, Terefenko and Paprotny [27], consumers' ability to purchase transactions is a crucial determinant of the adoption of cashless payment services. Wang et al. [25] indicated that compatibility has a significant impact on the adoption of online payment systems. When consumer-based organizational systems are perceived as being

compatible with the business' organizational experiences and values, they are more likely to be adopted. Liebana-Cabanillas et al. [26] stated that compatibility contains the consistency of innovation, with the experiences and values of an individual. Hence, we postulate that businesses are more likely to adopt cashless payment systems when they observe that cashless practices are compatible with their management practices, values, and experience. Therefore, we offer the following hypothesis:

**Hypothesis 1.** *Compatibility has a positive relationship with the adoption of cashless payment systems.*

### 2.2. Top Management Support

Top management support denotes the degree to which the top management realizes the significance of, and is involved in, the adoption of cashless payment systems [21,28]. Top management support is a crucial factor in the adoption of other technologies [29,30] such as mobile payment systems [31], supply chain management [32], and business intelligence systems [33]. The top management has the capabilities and technological resources to support the organizational adoption of technologies [29,30]. Top management provides important decision making for the organization, and they can enable a good environment to facilitate the adoption of cashless payments. Top management can provide sufficient resources and a creative vision for how technology adoption will facilitate business organizations. In one study, top management support was found to be significantly related to the firm's adoption or future usage of e-supply chain management [17]. In other studies, Pan and Pan [34] indicated that top management support has a significant impact on the adoption of construction robots, while Dubey et al. [35] found that top management support can encourage the diffusion of technology. Top management support also reflects on the success of technology adoption. Hence, the following hypothesis is offered:

**Hypothesis 2.** *Top management support has a positive relationship with the adoption of cashless payment systems.*

### 2.3. Firm Size

Different studies on technology adoption have used firm size as an organizational factor [17,29]. In the context of developed countries, Lowe et al. [36] indicated that firm size is important in defining the usage of audits, whereas Nnaji, Gambatese, Karakhan and Eseonu [37] documented that firm size is a significant predictor for the adoption of technology innovation. Compared to small-sized firms, larger-sized firms have more economies of scale that can support the hiring of a larger number of employees. Large firms are more capable of providing technology to facilitate the adoption of innovation. Siew et al. [29] indicated that firm size is a significant antecedent to the adoption of audit tools and techniques. As the significance of firm size is found in developed countries, we assume that firm size has a crucial influence on the adoption of a cashless payment system by businesses in Malaysia. As previous studies [34,38] have demonstrated that firm size influences the adoption of new technologies, we thus hypothesize:

**Hypothesis 3.** *Firm size is positively related to the adoption of a cashless payment system.*

### 2.4. Technology Competence

Business organizations have always leveraged the capabilities of information technology [39]. However, there is little understanding of how technology competence supports greater adoption of cashless payment systems. Technology competence signifies the internal technical resources of business organizations [40]. Meanwhile, previous studies have identified technological innovation as one of the most important factors in the business adoption of innovative electronic communications and transactions [41]. Since technological competence is closely related to innovative technologies such as cashless payment

systems [17], we strongly believe that technological competence is an important antecedent for this study. Hence, this hypothesis is offered:

**Hypothesis 4.** *Technological competence has a positive relationship with the adoption of the cashless payment system by businesses.*

### 2.5. Competitive Pressure

Competitive pressure is recognized as an important environmental factor [42]. Meanwhile, studies also found that a business organization adopts technological innovation to avoid the risk of competitive disadvantage [17]. Competitive pressure is commonly identified with travel organizations' adoption of e-commerce [43]. Mendi and Costamagna [44] indicated that the empirical evidence for the association between competitive pressure and the adoption of innovation is mixed. Ghosh et al. [45] pointed out that competitive pressure is an important antecedent of e-business adoption, while Hojnik and Ruzzier [46] found that competitive pressure is the strongest driver of process eco-innovation. Sin et al. [43] demonstrated that the higher the levels of competitive pressure on business organizations, the more likely they are to adopt internet business. Lin's [47] study found that competitive pressure has a significant relationship with business organizations' adoption of online supply chain management systems. In short, competitive pressure is more likely to lead to the adoption of cashless payment systems by businesses when they consider that non-adoption of cashless payment systems will lead to a disadvantage in society. On this basis, the following hypothesis is offered:

**Hypothesis 5.** *Competitive pressure has a positive relationship with the adoption of cashless payment systems by businesses.*

### 2.6. Critical Mass

Critical mass is an important environmental factor [17]. In the adoption of cashless payment systems, critical mass refers to the idea or starting point after which innovation adoption by an individual becomes self-sustaining [17]. A tipping point at the individual level can be reached if an individual has the awareness that a certain number of others have adopted a particular innovation. A study by Lallmahomed, Lallmahomed and Lallmahomed [48] shows that critical mass is related to externalities, which recommends that the value of the service rises with the number of clients. It implies that critical mass is the source for creating aggregate activities. Rhein [49] found that critical mass significantly contributes to the innovation adoption of businesses. It may be observed that when several individuals move to form a consistent minority, they can set up the desired critical mass, and their ideas can contribute to the adoption of organizational innovation. The rate of adoption will increase if more individuals from a framework observe that everyone is utilizing the adoption of innovation. The adoption of a cashless payment system is related to innovation. Sudan et al. [50] mentioned that perceptions of the critical mass of instant messaging significantly affect the adoption of innovation. The perceived lack of critical mass hinders businesses' adoption of innovative payment systems [51]. With regards to this, it might be expected that business organizations are more likely to adopt cashless payment systems when they identify that many individuals and businesses use this technological innovation. Hence, the following hypothesis is offered:

**Hypothesis 6.** *Critical mass has a positive relationship with the adoption of cashless payment systems by businesses.*

### 2.7. Information Intensity

Information intensity is an important tool in handling products with high information intensity [52]. The past literature has demonstrated that information intensity is an important predictor of the adoption of innovation information [17,52] that should be processed

by businesses along the value chain. High information intensity leads businesses to be more likely to adopt innovative information systems. Neirotti and Pesce [52] reported that information intensity can play a strategic role in business, and they highlighted that in the information age, it is utilized to attain a competitive advantage. Meanwhile, high information intensity provides benefits for improvement in business skills. For example, a location-based service provides users with skills and capacities in real-time positioning, and links them to useful information, such as traffic conditions, routes, and weather, that involve their businesses [53,54]. Coleman et al. [55] indicated that local activity, restaurants, and transportation create the most requested information. In highly information-intensive businesses, the adoption of cashless payment systems may be utilized to assist the business process. Along these lines, we have postulated that businesses are more likely to adopt cashless payment when they observe their information services to be useful and intensive. In turn, we propose the following hypothesis:

**Hypothesis 7.** *Information intensity has a positive relationship with the adoption of cashless payment systems by businesses.*

*2.8. Underpinning Theories for Cashless Payments*

Roger's [42] diffusion of innovation theory is adopted to predict factors influencing the adoption of cashless payment systems. This theory emphasizes innovation attributes [21,56] and the TOE framework or factors [17,21], such as compatibility (technological factor), firm size, top management support, critical mass (organizational factors), technology competence, competitive pressure, and information intensity (environmental factors) [21]. This research uses the technology, organization, and environment (TOE) framework/factors [16] to evaluate which factors provide an impact on the adoption of cashless payment systems. The diffusion of innovation theory can explain the spread of cashless payment when customers seek convenient transactions and businesses seek new profit benefits [1,3]. The significance of this theory in cashless payment systems relies upon how rapidly society is ready to adopt a cashless payment method through the innovation processes. For this study, we have developed an integrated model of businesses' adoption of cashless payment systems. Figure 1 shows the integrated model.

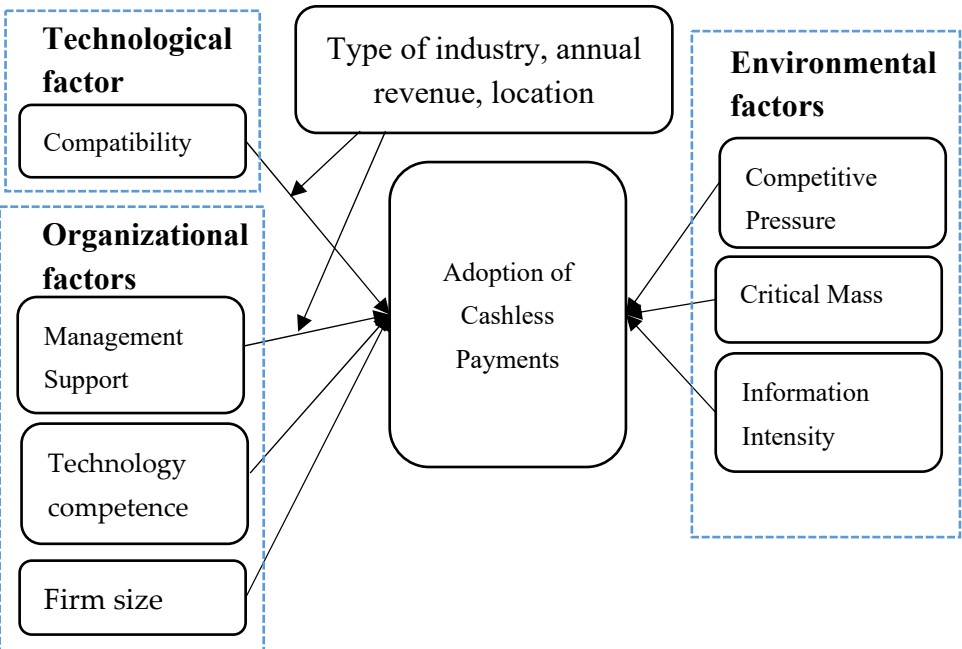

**Figure 1.** Conceptual model.

## 3. Research Methodology

### 3.1. Data Collection and Sample

Malaysia's migration towards a cashless society is one of the country's main thrusts towards embracing the digital economy. This is inevitable in the coming era of Industrial Revolution 4.0. To understand the pull and push factors towards achieving cashless status, we collected feedback from the various stakeholders that would be driving this initiative forward. This feedback was obtained through one-to-one personal discussions with commercial bank representatives, the regulator (Bank Negara Malaysia), and Fintech firms providing cashless payment systems. From these discussions, we identified that management support, compatibility with business information technology infrastructure, firm size, critical mass, technology competence, competitive pressure, and information intensity are the potential factors for the adoption of cashless payment systems by businesses, in the context of stakeholders in Malaysia. After the discussion, we designed the questionnaires and surveyed 200 businesses/organizations to understand the factors that motivate their adoption of the various cashless payment methods. To understand the perspective of local businesses, a survey was conducted among businesses based in the Klang Valley in Malaysia (which encompasses the capital city, Kuala Lumpur). As the respondents were based within the capital city, these businesses were deemed to be able to benefit the most from the infrastructure that supports cashless payments and that, therefore, would give us feedback from those at the forefront of this movement.

A total of 500 businesses were approached for data collection, whereby 200 completed questionnaires were returned and used for data analysis, giving a response rate of 40%. Usually, businesses are busy with their respective meetings, and other scheduled work, thus some respondents may not have been able to make time to complete the questionnaires. The surveyed organizations and prospective respondents were purposively selected and physically distributed, as these procedures enabled us to access a sufficient number and variety of respondents for this study. Prior to sending out the questionnaires, ethics approval was obtained from the University of Malaya, which was included in the survey information disclosed to the selected organizations to get permission for the data collection. Two sets of questionnaires, in English and Bahasa Malaysia, with a plain cover letter about the statement related to the main objective of this survey, were distributed by trained enumerators allowing for face-to-face interaction with the management or senior employee of each business operation. The questionnaires were pre-tested by five experts in this area of study who suggested changes to several items. The survey took around 15 to 20 min to be completed by the respondents. The participants were ensured that this study would be solely used for academic purposes, and their participation was voluntary and anonymous. The questionnaire was divided into two sections. In the Section 1, questions were related to the demographic information and types of industry, size of the organization, number of years in operation, gross annual revenue, location of operation, current and future payment systems, as well as riskiness of payment methods. The Section 2 contained several items to measure the adoption of cashless payment systems for business, technology competence, compatibility, competitive pressure, critical mass, information intensity, management support, and the size of the firm.

For this study, G-Power version 3.1 was employed to test the appropriate sample size. Based on the effect size of 0.15, the G-Power tool recommends a minimum sample size of 165 for the existing model with eight variables. The G-Power test result shows a significant value of 0.05, generating a strength of 0.99 (which is above 0.80), indicating a satisfactory level of sample power [57]. For the SmartPLS analysis, Reinartz et al. [58] suggested a minimum sample of 100. Accordingly, this study collected data from 200 businesses, which is above the minimum required sample size.

### 3.2. Measurement Operationalization

The survey instruments for independent and dependent variables were evaluated using a six-point Likert scale from 1 (strongly disagree) to 6 (strongly agree). To evaluate

the adoption of cashless payment, five items were modified from [24,59]. Three items were adapted from [17,24] to evaluate the compatibility. These items were important as compatibility constitutes an organizer, and it is crucial for the consistency between the adoption of cashless payment systems, and their values and experiences. Based on Wong et [21] and Koh's [60] study, three items were adapted to assess top management support. Firm size is perhaps the most crucial illustrative factor for businesses in their adoption of cashless payment systems. To measure the firm size, three items were modified from Wang et al. [17] and Andries and Stephan [61]. Critical mass was evaluated in this study using three items adapted from Wang et al. [17], Van Slyke et al. [62], Lallmahomed et al. [48], and Zhou and Li [63]. Three items were modified from Mao et al. [64] to estimate information intensity. To evaluate technical competence, three items were adapted from Wang et al. [17] and Deligianni et al. [41]; these items cover the tangible and intangible resources of the organizations, such as technological expertise and production facilities, which enable businesses to produce cashless products with specific features. Based on Wong et al. [21] and Siew et al. [29], three items were adapted to measure competitive pressure, which is perceived by the incumbent, and the perception of competition is connected between newcomers in the market and stakeholders.

### 3.3. Common Method Bias Test

As structured questionnaires were used for collecting data, common method bias threatened the validity of the result. To address this issue, we used Harman's single-factor test to determine the seriousness of the issue [65]. We also conducted a factor analysis to measure the level of the negative influence of common method bias. The results showed that all the items loading were significant. However, the result explained less than 50% of the total variance, indicating a possible common method bias [66]. Fortunately, the correlation coefficients shown in convergent validity Table below are less than 0.90, indicating that the issue of common method bias is not severe [67].

### 3.4. The Analysis Tool

The Statistical Package for the Social Sciences (SPSS 23) and SmartPLS 3.0 were used to run the measurement and structural model analysis. Three-step procedures were employed to examine the characteristics of respondents and the conceptual model of this study. The first step involved the descriptive analysis of respondents' demographic profile, while the second step assessed the convergent and discriminant validity. The third step measured the hypothesized relationship between the exogenous and endogenous constructs.

## 4. Data Analysis and Results

### 4.1. Profiles of Respondents and Their Use of Cashless Payment Systems

In terms of the number of years in operation, the surveyed businesses had been in operation between 3 to 24 years. As for the number of employees, out of 200 participants, 79% of the businesses had less than 10 employees. Meanwhile, businesses with revenues of less than MYR100,000 (about USD25,000) and between MYR100,101–500,000 (about USD25,000–125,000), made up 42% and 37% of the respondents, respectively. As for location, 73% of the businesses were located in shop lots. In a sense, the sample represents small businesses typically involved in the retail sector. The viewpoints of this group will be of greater interest than larger businesses that are more likely to embrace cashless payments due to the larger size of their transactions, which makes payment in cash cumbersome. Small business payments are typical of smaller amounts but with higher transaction frequency.

Figure 2 shows the different types of payment systems. We found that most of the businesses (90%) indicate that they often accept cash as a payment method. This finding stands in stark contrast to the number of businesses that accept the cashless form of payment. Only 36% of the businesses indicate that they accept credit cards, and slightly less than that (34%) accept debit cards. The number is much lower for other types of cashless payment methods. Only 28% of businesses accept payments online or through the internet,



16% accept payments through mobile phones, 9% accept payments using prepaid cards (i.e., Touch 'n Go), and only 6% of businesses accept cheques and cash deposit machines (CDM). These findings suggest that many businesses in this country have yet to fully embrace cashless payment systems and still rely heavily on cash for settling transactions.

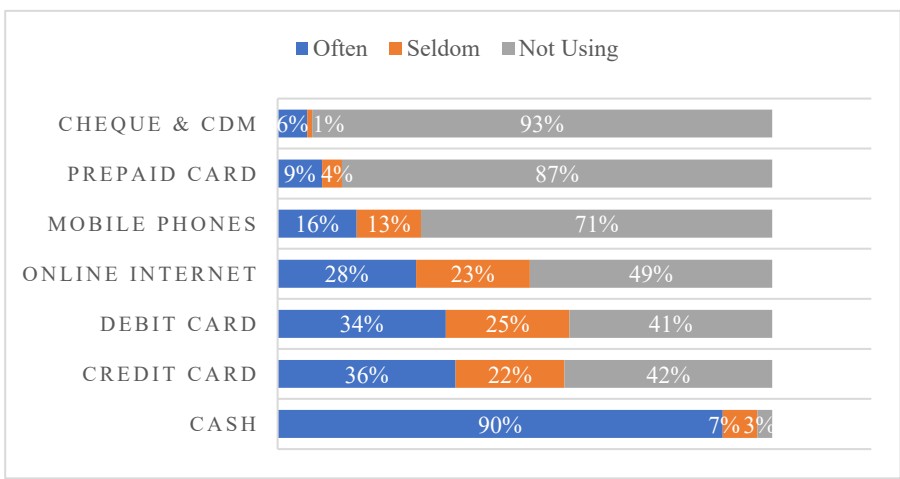

**Figure 2.** Existing payment systems.

In terms of future use of cashless payment systems, more businesses indicate that they are willing to adopt cashless payment. Out of the nine cashless payment systems, almost half of the businesses responded that they are using credit (53%) and debit cards (51%), respectively. However, more than half of them are using Bitcoin, mobile credit, Apple/Samsung pay, loyalty points and prepaid cards (i.e., 81%, 73%, 70%, 61%, and 55%, respectively). Additionally, the number looks optimistic for online internet banking and mobile payments. The findings reveal that about 38% of respondents are using online internet banking, whilst 22% of respondents have made payments via mobile phones. However, the numbers for mobile payments are rather low, considering the widespread use of smartphones among customers in urban areas. Strategies to increase the acceptance of mobile payments should be explored to take advantage of this dynamic.

Assessing the perceived riskiness of cash and cashless payment systems, the result of our business survey suggests that the four cashless payment systems with the highest risks are Bitcoin (71%), Apple/Samsung Pay (63%), mobile phones (53%) and mobile credit (52%). On the other hand, the least risky payment systems include prepaid cards (26%), debit cards (33%), and loyalty points (39%). It can be said that prepaid cards such as Touch 'n Go are perceived to be as safe as cash. Meanwhile, both credit cards and online internet banking were perceived to have medium risks (Figure 3). This finding perhaps explains why mobile payment acceptance is rather low among businesses. Steps need to be taken to improve the safety of mobile payments by investing in appropriate technologies to mitigate the risk of fraud.

*4.2. Normality and Multicollinearity Test*

Using the multivariate statistical test, this study's preliminary analysis includes data normality and a multicollinearity test. The result of the Kolmogorov–Smirnov one-sample test is shown in Table 1 for the normality of data distribution. The result indicates that the two-tailed asymptotic significance is less than 0.05, which signifies that the data is not normally distributed in this study. To check the multicollinearity issue, we have conducted collinearity statistics (VIF) using the SmartPLS analysis. We found that all VIFs were less than 3.3, indicating that there is no multicollinearity problem because, according to Kock [68], only VIFs greater than 3.3 indicate that there are high correlations and multicollinearity problems.

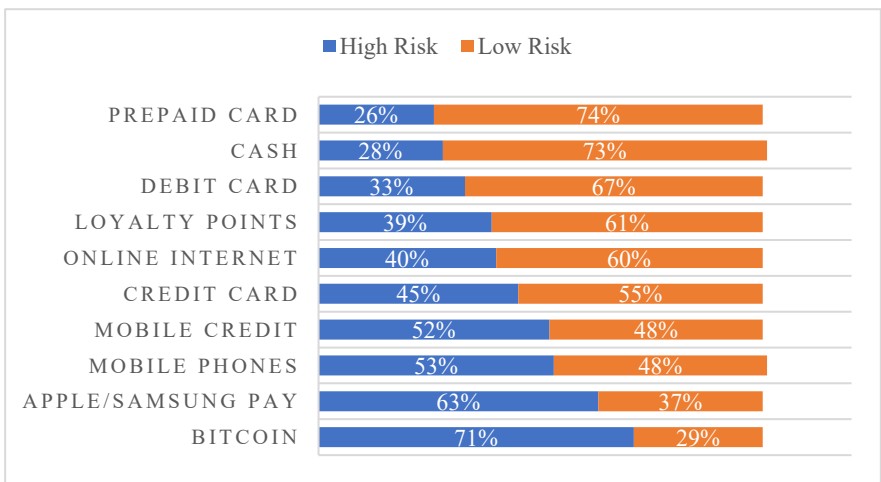

**Figure 3.** Security of cashless payment systems.

**Table 1.** One-sample Kolmogorov–Smirnov test.

| Code | VIF | Normal Parameters [a,b] | | Most Extreme Differences | | | Kolmogorov–Smirnov Z | Asymp. Sig. (2-Tailed) |
| | | Men | Std. D | Absolute | Positive | Negative | | |
|---|---|---|---|---|---|---|---|---|
| B11.1 | 2.000 | 4.34 | 1.291 | 0.146 | 0.146 | −0.145 | 2.423 | 0.000 |
| B11.2 | 2.558 | 4.78 | 1.253 | 0.189 | 0.189 | −0.143 | 2.544 | 0.000 |
| B11.3 | 2.548 | 4.65 | 1.216 | 0.180 | 0.180 | −0.161 | 2.623 | 0.000 |
| B12.1 | 2.452 | 5.01 | 1.178 | 0.167 | 0.147 | −0.139 | 2.268 | 0.000 |
| B12.2 | 2.098 | 5.05 | 1.174 | 0.196 | 0.196 | −0.164 | 2.244 | 0.000 |
| B12.3 | 2.042 | 5.03 | 1.163 | 0.166 | 0.146 | −0.167 | 2.504 | 0.000 |
| B13.1 | 1.653 | 4.40 | 1.201 | 0.187 | 0.187 | −0.201 | 2.786 | 0.000 |
| B13.2 | 1.682 | 4.11 | 1.322 | 0.167 | 0.160 | −0.243 | 3.110 | 0.000 |
| B13.3 | 1.682 | 4.04 | 1.197 | 0.179 | 0.164 | −0.168 | 3.215 | 0.000 |
| B14.1 | 1.778 | 4.48 | 1.203 | 0.188 | 0.188 | −0.157 | 2.541 | 0.000 |
| B14.2 | 2.568 | 3.17 | 1.235 | 0.175 | 0.175 | −0.269 | 3.671 | 0.000 |
| B14.3 | 1.887 | 3.84 | 1.050 | 0.171 | 0.171 | −0.146 | 2.246 | 0.000 |
| B15.1 | 1.453 | 3.95 | 1.113 | 0.211 | 0.211 | −0.162 | 3.508 | 0.000 |
| B15.2 | 1.385 | 3.22 | 1.047 | 0.214 | 0.214 | −0.193 | 3.711 | 0.000 |
| B15.3 | 1.346 | 3.02 | 1.070 | 0.158 | 0.158 | −0.262 | 3.675 | 0.000 |
| B16.1 | 1.255 | 4.00 | 1.220 | 0.209 | 0.209 | −0.188 | 2.802 | 0.000 |
| B16.2 | 1.849 | 4.09 | 1.076 | 0.157 | 0.159 | −0.243 | 2.876 | 0.000 |
| B16.3 | 1.536 | 4.01 | 1.248 | 0.178 | 0.164 | −0.242 | 3.721 | 0.000 |
| B17.1 | 1.130 | 3.64 | 1.414 | 0.243 | 0.243 | −0.212 | 2.967 | 0.000 |
| B17.2 | 1.659 | 3.53 | 1.334 | 0.206 | 0.206 | −0.153 | 2.435 | 0.000 |
| B17.3 | 1.667 | 3.84 | 1.291 | 0.176 | 0.176 | −0.783 | 2.243 | 0.000 |
| B9.1 | 1.959 | 3.32 | 1.210 | 0.227 | 0.227 | −0.232 | 3.124 | 0.000 |
| B9.2 | 2.104 | 3.19 | 1.196 | 0.233 | 0.233 | −0.240 | 3.418 | 0.000 |
| B9.3 | 1.821 | 3.72 | 1.436 | 0.205 | 0.205 | −0.186 | 2.728 | 0.000 |
| B9.4 | 2.153 | 3.62 | 1.150 | 0.213 | 0.213 | −0.189 | 2.689 | 0.000 |
| B9.5 | 1.763 | 3.38 | 1.082 | 0.204 | 0.204 | −0.168 | 2.783 | 0.000 |

Note: B11.1–B12.3 = Compatibility, B12.1–B12.3 = Management support, B13.1–B13.3 = Firm size, B14.1–B14.3 = Technology competence, B15.1–B15.3 = Competitive pressure, B16.1–B16.3 = Critical mass, B17.1–B17.3 = Information intensity, B9.1–B9.3 = Adoption of cashless payments, VIF = Variance Inflation Facto/Collinearity Statistics. [a] Test distribution is Normal. [b] Calculated from data.

### 4.3. Measurement Model Analysis

To assess data consistency, Dijkstra–Henseler's rhoA was employed in the measurement of construct reliability. Dijkstra and Henseler [69] suggested that composite reliability (CR), Cronbach's alpha (α), and rhoA are required to meet in the range between 0.70 and 0.95. Table 2 showed that all of these values attained their satisfactory levels, thus indicating that the measurement model has internal consistency. Convergent validity was achieved,

as all the variables' average variance extracted (AVE) was greater than 0.50 [70]. For the reliability of factor loading, Gudergan et al. [71] and Sarstedt et al. [72] suggested that loading higher than 0.70 is good but if AVE has not attained a satisfactory level, then the value between 0.40 and 0.70 may be dropped. The AVE value in this study was between 0.513 and 0.812, indicating that they meet the convergent validity criteria. The results of the measurement model are shown in Figure 4.

**Table 2.** Convergent Validity.

| Variables and Items | FL | CA | rhoA | CR | AVE |
|---|---|---|---|---|---|
| **Compatibility (CO)** | | 0.870 | 0.871 | 0.920 | 0.793 |
| (11.1) The changes introduced by cashless payment systems are consistent with our business's existing beliefs/values. | 0.876 | | | | |
| (11.2) A cashless payment system is compatible with our business's existing information technology infrastructure. | 0.896 | | | | |
| (11.3) The changes introduced by the cashless payment system are consistent with our business's existing operations. | 0.900 | | | | |
| **Management Support (MS)** | | 0.858 | 0.890 | 0.912 | 0.776 |
| (12.1) Our top management is likely to invest in cashless payment systems. | 0.914 | | | | |
| (12.2) Our top management is willing to take the risks involved in the adoption of cashless payment systems. | 0.839 | | | | |
| (12.3) Our top management is keen to adopt cashless payment systems to gain a competitive advantage. | 0.887 | | | | |
| **Firm Size (FS)** | | 0.778 | 0.884 | 0.896 | 0.812 |
| (13.1) The capital of my business is high compared to the business industry in general. | *Drop 0.345* | | | | |
| (13.2) The revenue of my business is high compared to the business industry in general. | 0.943 | | | | |
| (13.3) The number of employees at my business is high compared to the business industry in general. | 0.857 | | | | |
| **Critical Mass (CM)** | | 0.755 | 0.671 | 0.814 | 0.596 |
| (16.1) Most of my business customers use smartphones. | 0.717 | | | | |
| (16.2) Most of my business customers download cashless payment systems-related applications via smartphones. | 0.882 | | | | |
| (16.3) Most of my businesses use cashless payment systems via smartphones. | 0.704 | | | | |
| **Technology Competence (TC)** | | 0.824 | 0.887 | 0.894 | 0.739 |
| (14.1) Our current ICT infrastructure can support cashless payment systems applications without much investment or restructuring. | 0.812 | | | | |
| (14.2) Our business is dedicated to ensuring that employees are familiar with cashless payment systems-related technology. | 0.935 | | | | |
| (14.3) Our employees demonstrate a high level of knowledge for cashless payment systems applications. | 0.826 | | | | |
| **Competitive Pressure (CP)** | | 0.791 | 0.705 | 0.865 | 0.762 |
| (15.1) Our business faced competitive pressure to implement cashless payment systems. | *Drop 0.401* | | | | |
| (15.2) Our competitors who implemented cashless payments systems early enough have gained a competitive advantage. | 0.896 | | | | |
| (15.3) We believe that we might lose customers to our competitors if we had not adopted cashless payment systems. | 0.849 | | | | |
| **Information Intensity (II)** | | 0.776 | 0.788 | 0.733 | 0.513 |
| (17.1) Customers in our business industry generally require a lot of information before purchasing products/services. | 0.868 | | | | |
| (17.2) Products/services in our business industry are complex and hard to understand. | 0.724 | | | | |
| (17.3) The booking process in our business industry is generally complex. | 0.451 | | | | |
| **Adoption of Cashless Payments (ACP)** | | 0.836 | 0.840 | 0.884 | 0.605 |
| (9.1) We expect cashless payment systems to help increase market share. | 0.746 | | | | |
| (9.2) We expect cashless payment systems to help speed up the transaction process. | 0.776 | | | | |
| (9.3) We expect cashless payment systems to help lower costs. | 0.786 | | | | |
| (9.4) We expect cashless payment systems to help enhance interaction with third parties. | 0.830 | | | | |
| (9.5) We expect cashless payment systems to improve transparency between stakeholders. | 0.747 | | | | |

Note: FL = Factor loading, CA = Cronbach's Alpha, CR = Composite Reliability, AVE = Average Variance Extracted.

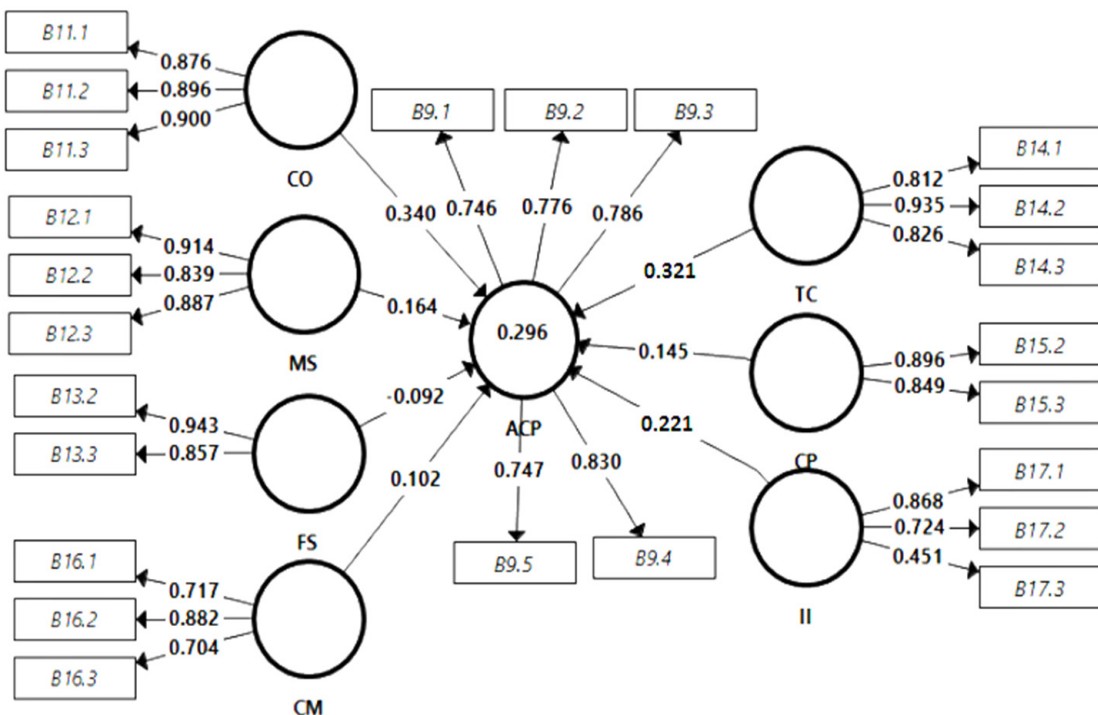

**Figure 4.** Measurement model.

Several procedures were employed for the discriminant validity (Table 3) criterion. First, we used Fornell–Larcker's [70] criterion. The result indicated that the square root of AVE was higher (italic and bold values at the top of the columns) than the correlation coefficients. Second, the heterotrait–monotrait ratio (HTMT) [73] was used to examine the discriminant validity. Figure 5 shows that all HTMT ratios of correlations are below 0.70, which is less than 0.85, and meets the discriminant validity criterion. According to Hair et al. [74], a factor with an HTML ratio of less than 0.85 has discriminant validity. Several scholars argued that the HTMT ratio must not exceed 0.90, or a minimum threshold of 0.85 [73].

**Table 3.** Discriminant Validity (Fornell–Larcker Criterion).

|  | ACP | CM | CO | CP | FS | II | MS | TC |
|---|---|---|---|---|---|---|---|---|
| ACP | *0.778* | | | | | | | |
| CM | 0.348 | *0.772* | | | | | | |
| CO | 0.489 | 0.503 | *0.891* | | | | | |
| CP | 0.218 | 0.254 | 0.118 | *0.873* | | | | |
| FS | 0.093 | 0.309 | 0.184 | 0.277 | *0.901* | | | |
| II | 0.068 | 0.280 | 0.146 | 0.140 | 0.176 | *0.702* | | |
| MS | 0.395 | 0.375 | 0.539 | 0.188 | 0.300 | 0.018 | *0.881* | |
| TC | 0.365 | 0.514 | 0.593 | 0.240 | 0.271 | 0.188 | 0.514 | *0.860* |

Note: Adoption of Cashless Payments (ACP), Critical Mass (CM), Compatibility (CO), Competitive Pressure (CP), Firm Size (FS), Information Intensity (II), Management Support (MS), Technology Competence (TC).

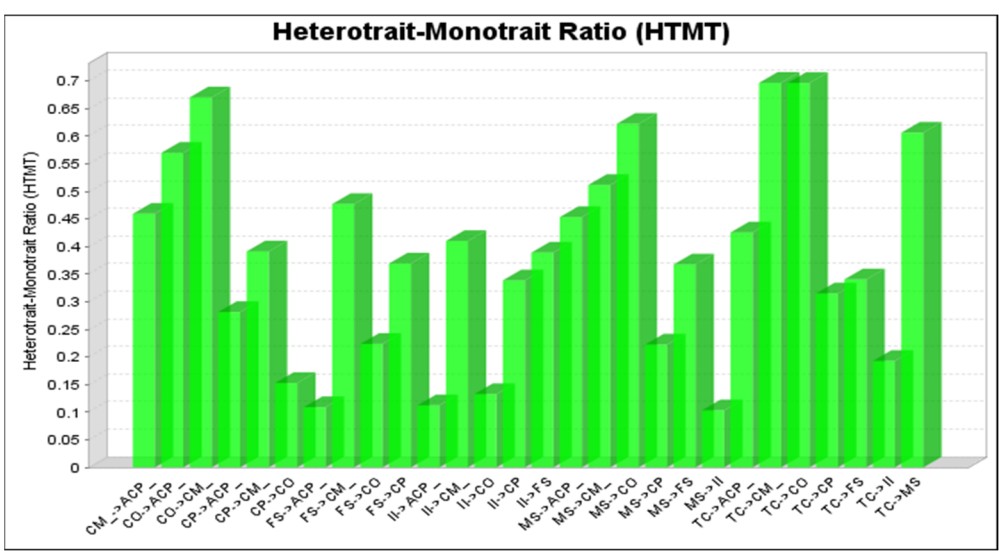

**Figure 5.** Heterotrait-Monotrait Ratio (HTMT).

*4.4. Structural Model Analysis*

For the validation of the hypotheses' relationships, we employed SmartPLS 3.0 [75]. Here, we linked constructs through a set of paths in the structural model, which reflect the hypotheses. The relationships between variables can capture direct effects; for instance, the path between the independent variables and dependent variable in Figure 6 exemplifies a direct relationship. Chin and Dibbern [76] reported that this method is useful for exploratory studies. Following Hair et al. [74], we have observed indicators of a good fit of the model, such as predictive relevance ($Q^2$), path coefficient ($\beta$), and their coefficient of determination ($R^2$) or confidence intervals for the evaluation of the structural model. The Q-square values higher than 0.35 indicate that the exogenous variable has strong predictive relevance for a given endogenous variable [77].

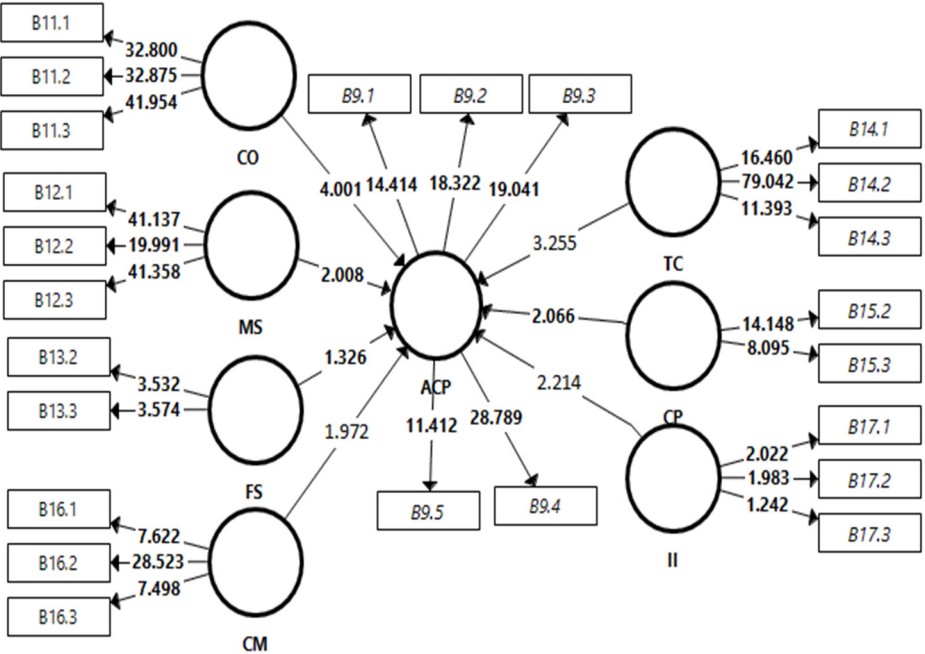

**Figure 6.** Structural model.

Table 4 showed the predictive relevance of Stone–Geisser's $Q^2$ value [78] The findings revealed that the structural model has a Q-square threshold between 0.065 and 0.521,

indicating good results. Hence, the independent variables were highly relevant to the dependent variable. Furthermore, the Q-square values are higher than zero, which indicates that the model has predictive validity [57]. R-square showed the combined results of independent variables on the dependent variable. Hair et al. [79] pointed out that it is difficult to ascertain accurate rules for acceptable R-square values. Hence, an R-square threshold of 29.6% can be considered as a good result. In terms of the effect size ($f^2$), the f2 value of 0.02 denotes a small effect size, 0.15 indicates medium, and 0.35 signifies a large effect size [80]. We found that the adoption of cashless payment has a high effect size of 0.387; critical mass (0.223), competitive pressure (0.209), firm size (0.200), and information intensity (0.226) have a medium effect size; and compatibility has a small effect size of 0.023.

**Table 4.** Stone–Geisser's $Q^2$ value.

| Constructs | SSO | SSE | $Q^2$ (=1-SSE/SSO) | $f^2$ |
|---|---|---|---|---|
| Adoption of Cashless Payments (ACP) | 1000.000 | 599.800 | 0.400 | 0.387 |
| Critical Mass (CM) | 600.000 | 463.596 | 0.227 | 0.223 |
| Compatibility (CO) | 600.000 | 287.449 | 0.521 | 0.023 |
| Competitive Pressure (CP) | 400.000 | 290.235 | 0.274 | 0.209 |
| Firm Size (FS) | 400.000 | 255.883 | 0.360 | 0.200 |
| Information Intensity (II) | 600.000 | 639.048 | 0.065 | 0.226 |
| Management Support (MS) | 600.000 | 299.776 | 0.500 | 0.101 |
| Technology Competence (TC) | 600.000 | 329.339 | 0.451 | |

For the expected significance level of hypotheses, a t-value of 2.326 assumes a significant difference at 0.01, while a t-value of 1.645 assumes a significant difference at 0.05 [81]. In this regard, we used a bootstrapping method with 5000 resamples, which enabled the assessment of the path coefficients [74]. Figure 6 shows the results of the structural model by using the bootstrapping confidence intervals of standardized regression coefficients. The test checks the strength of the structural path in determining the goodness-of-fit of the proposed business adoption of the cashless payment systems model. The standardized root means that the square residual (SRMR) value required in determining the goodness-of-fit for both measurement and structural model should be less than 0.0813 [77]. This study's SRMR value for the measurement is 0.074, and the structural model is 0.080. These results suggest that the structural model has satisfactory predictive relevance for the dependent variable.

Table 5 shows the significant path coefficient relationships between exogenous and endogenous constructs. The findings reveal that compatibility and adoption of cashless payments are highly significant and positively linked ($\beta$ = 0.340, $t$ = 4.001, $p < 0.01$), and this lends support to hypothesis 1. Similarly, technology competence and the adoption of cashless payments were highly significant and positively linked ($\beta$ = 0.321, $t$ = 3.867, $p < 0.01$), and this provides support to hypothesis 5. The findings also revealed that top management support ($\beta$ = 0.164, $t$ = 2.008, $p < 0.05$), critical mass ($\beta$ = 0.102, $t$ = 1.924, $p < 0.05$), competitive pressure ($\beta$ = 0.145, $t$ = 2.066, $p < 0.05$), and information intensity ($\beta$ = 0.221, $t$ = 2.302, $p < 0.05$) are significantly and positively related to the adoption of cashless payments. Thus, hypotheses 2, 4, 6, and 7 were supported. However, firm size and the adoption of cashless payments are not significantly related ($\beta$ = −0.092, $t$ = 1.326, $p > 0.05$) and therefore hypothesis 3 is not supported.

**Table 5.** Path Coefficients.

| No | Hypothesis | Coefficient | Std. | *t*-Value | 2.50% | 97.50% | Decision |
|---|---|---|---|---|---|---|---|
| 1 | CO → ACP | 0.340 | 0.085 | 4.001 ** | 0.187 | 0.512 | Supported |
| 2 | MS → ACP | 0.164 | 0.082 | 2.008 * | 0.106 | 0.309 | Supported |
| 3 | FS → ACP | −0.092 | 0.069 | 1.326 | −0.208 | 0.077 | Not Supported |
| 4 | CM → ACP | 0.102 | 0.063 | 1.924 * | 0.185 | 0.245 | Supported |
| 5 | TC → ACP | 0.321 | 0.083 | 3.867 ** | 0.134 | 0.186 | Supported |
| 6 | CP → ACP | 0.145 | 0.070 | 2.066 * | 0.016 | 0.297 | Supported |
| 7 | II → ACP | 0.221 | 0.096 | 2.302 * | 0.214 | 0.155 | Supported |

Note: *t*-value $\geq 2.326$ considers significant level at ** $p < 0.01$ and *t*-value $\geq 1.645$ considers significant level at
* $p < 0.05$.

To identify factors that moderate the predictors of adopting cashless payments for business, this study considered a few characteristics of the business, such as type of industry, annual revenue, and location of the firm. Table 6 reports that the compatibility of the firm has a significant positive effect on the intention to adopt a cashless payment system. The findings of moderation analysis indicate that the type of the industry can significantly influence the relationship between technological compatibility and the adoption of a cashless payment system. However, revenue and location do not have a moderating effect on that relationship. It is also found that the location of the business does moderate the relationship between top management support and the adoption of the cashless payment system. This finding suggests that a business that is in an area where most people like to use cashless payments and has support from the top management would be more willing to adopt a cashless payment system compared to a business that is in an area where most people use cash.

**Table 6.** Moderating effect.

| No. | Variable | β | SE | *t* | Sig. |
|---|---|---|---|---|---|
| 8 | Compatibility | 0.510 ** | 0.148 | 3.257 | 0.001 |
| 8a | Moderating COMP_Type of industry | 0.140 * | 0.012 | 2.118 | 0.035 |
| 8b | Moderating COMP_Revenue | 0.098 | 0.027 | 1.167 | 0.245 |
| 8c | Moderating COMP_Location | 0.141 | 0.029 | 1.331 | 0.185 |
| 9 | Top Management Support | 0.277 * | 0.124 | 2.021 | 0.045 |
| 9a | Moderating TMS_Location | 0.252 ** | 0.026 | 2.536 | 0.012 |
| 9b | Moderating TMS_Type of industry | −0.153 | 0.028 | −0.153 | 0.196 |
| 9c | Moderating TMS_Revenue | 0.058 | 0.024 | 0.728 | 0.467 |

Note: Dependent Variable: adoption of cashless payment system, COMP = Compatibility, TMS = Top Management Support, Significant level at * $p < 0.05$. ** $p < 0.01$.

## 5. Discussion, Cashless Payment System, and Open Innovation

### 5.1. Cashless Payment System

This study examines the factors affecting businesses' adoption of cashless payment systems through the theoretical lens of the TOE framework. The results indicate that compatibility and technology competence is closely linked with the adoption of cashless payment systems. These findings suggest that businesses that have the existing infrastructure to accommodate cashless payments, such as internet banking facilities, would be more willing to adopt a cashless payment system. Similar findings were found by Wang et al. [17], who examined RFID adoption in the manufacturing industry, whereby existing infrastructure, such as computer networks, helped accelerate the take-up of the technology. Meanwhile, technological competence is important for the adoption of cashless payment systems by businesses. Similarly, Wang et al. [17] found that technological competence is an important prerequisite for the adoption of MHRS. Interestingly, firm size is not significantly associated with the adoption of cashless payment systems. These findings are contradictory to Wang et al. [17], who found that firm size is important for business adoption. However,

Wang et al.'s [17] findings differ from Lin's [47], who indicated that firm size is not related to the adoption of electronic supply chain management systems. Perhaps the reason behind the insignificant effect of firm size is that big firms need to invest more capital, compared to small firms, into developing the infrastructure required to adopt a cashless payment system. An alternative explanation could be that the investment required to prepare the system is not as prohibitive as many had expected. Thus, both small and large firms are able to adopt the technology. The more important question facing these firms is their willingness to adopt cashless payment systems. Meanwhile, the regulator and system providers need to educate businesses to overcome the problems that they face in terms of developing supportive infrastructure to increase the adoption of the cashless payment system.

This study's result also indicates that top management support is crucial for the adoption of cashless payment systems. These findings differ from Soliman and Janz [82] and Wang et al. [17], who found that top management support is not closely related to internet-based inter-organizational information systems and hotels' adoption of MHRS. One probable explanation for this finding is that the systems are highly linked with the collection of revenue for businesses. Choosing the wrong payment system can have an adverse effect on their revenue. Hence, top management support is crucial in the adoption of payment systems. Meanwhile, critical mass is found to be significant, which is similar to the result of Mallat and Tuunainen's [83] study of business adoption of mobile payment systems. More businesses are willing to join the "bandwagon" of cashless payment systems when they see others doing so and reaping the rewards of their adoption. In another finding, information intensity is crucial for the adoption of cashless payment systems. However, it differs from the findings in Wang et al.'s [17] study of hotels' adoption of mobile reservation systems. This study additionally found that competitive pressure is influenced by the business environment that they operate. These findings are related to Wang et al.'s [17] study of the environmental features of competitive pressure that may influence hotels' adoption. These types of factors that fit in numerous perceptions, particularly in the adoption of cashless payment systems, are significant for moving businesses towards cashless payment systems and in terms of the combinations of these theoretically grounded variables.

*5.2. Cashless Payment System and Open Innovation*

The current adoption of an e-payment service, which is also known as a cashless payment system in Malaysia, is still small, and many challenges are yet to be addressed. Like many others, Kilay, Simamora and Putra [6] indicate that open innovations and solutions can play a significant role in accelerating the digitization of business services. The results of our study reveal that compatibility and technology competence have a significant impact on the adoption of cashless payment systems, and businesses may be deterred from adopting cashless payments due to incompatibility with their business operations. The coupling processes strategy of open innovation can be used to address the issue of compatibility and technological competencies of the businesses, which ultimately can accelerate the adoption of cashless payment systems. A previous study by Kilay, Simamora and Putra [6] suggested that open innovation significantly improves the business processes of micro, small, and medium Enterprises (MSMEs) in Indonesia.

In terms of competitive pressure, it has been shown that open innovation has a significant impact on developing the innovation capacity in businesses, helping them to achieve competitive advantage through providing faster, cost-saving, and efficient solutions to stakeholders. Applying open innovation may accelerate businesses' innovation process, including a cashless payments system and reducing competitive pressure to achieve sustainable growth in the business.

Top management support, firm critical mass, and information intensity are necessary to enable a good environment to facilitate the adoption of cashless payments. However, there is a reluctance from top management to adopt cashless payment systems as it requires investment in developing infrastructure, providing training to employees, as well not being

able to see the clear benefits or value of adopting it. Open innovation adoption enhances the growth of knowledge and a strong external network in businesses, which ultimately encourages top management to be more proactive in increasing its innovativeness. A previous study by Singh, Gupta, Busso and Kamboj [84], suggested that top management knowledge-creating practices enhance open innovation and firm performance. Businesses should use open innovation not only for the commercialization of their products, but rather for research and development to improve the payment mechanism, such as with a cashless payment system, to enhance its performance and profitability. The concept of open innovation helps to develop efficient and cost-saving solutions for both businesses and consumers. Hence, open innovation is recommended to address the various challenges that pose an obstacle to the adoption of the cashless payment system for businesses in Malaysia.

## 6. Implications

This study theoretically adds to the limited literature on the adoption of cashless payment systems for businesses in Malaysia. The research gap in understanding the key factors influencing the adoption of cashless payment systems is met by this study. We successfully explored the factors influencing the adoption of cashless payment systems, ensuring the acceptability of suggestions from the previous literature. The findings of this study indicated that the adoption of cashless payment systems can serve as an effective theory to study the subject matter. Business organizations should emphasize the facilities and better performance of cashless payment systems in their marketing campaigns. It can be achieved by promoting the advantages of the cashless payment system, for instance, frictionless payments and its time-saving nature. It is also necessary to take the initiative to address people's familiarity with the adoption of cashless payment systems. One way this can be performed is by highlighting to people the benefits of adopting cashless payment systems in their retail sectors. Cashless payment businesses should furnish periodical updates with detailed clarifications about the safety procedures in place to reassure clients when they use cashless payment systems. Cashless payment system developers should concentrate on the user interface, perhaps in terms of minimizing the number of steps required to complete cashless transactions and making the process easy to understand. In terms of risk in cashless payment systems, business developers need to focus on security aspects, for example, incorporating biometrics or authentication to prevent hacking. Businesses can play a crucial role by confirming that their retail sectors support cashless payment systems. They can encourage clients to utilize cashless payment systems by providing rewards, cash-back, and loyalty points. Businesses may also provide skilled staff to help clients carry out cashless payment transactions.

Policymakers are aware that banks have an advantage over non-bank players in the cashless payment system, and they have strived to provide a much more level playing field. The competition will hopefully lead to better products and choices for clients, and perhaps lower costs. However, opening up Malaysia's emerging market to large international service providers, before local players are better established, generates the risk that international competition will wipe them out completely. Tencent's launch of WeChat pay in the country is an experiment that regulators should closely monitor. Market dominance by a large payment company may put the risk of the payment system at the whim of external players. Less support for local Fintech players may lead them to relocate to other regional countries, as has been the case with some local tech startups. This will be a loss to Malaysia in the long term.

There could be other proposals that would help spur the cashless movement, such as encouraging government services to go cashless, especially frontline services provided by the Home Ministry. Cash is a public good, and it is free at the point of use. Cash handling costs get dispersed in society. E-payment costs, on the other hand, are borne by users. If another means of payment is forced upon society, then policymakers should ensure that a free means of payment exists in the chain. Going exhaustively cashless can be risky to the society, and an alternative payment system may be required, such as digital, gold, or other

countries' currencies, such as USD or euro. If this cashless movement is large enough, then there may arise issues in terms of control over monetary policy. Free market economics dictate that the markets, if left on their own, will reach the most efficient solution towards any problem. This view can also be applied to the whole cashless movement. However, we believe that, for a developing country such as Malaysia, there is an important role for the regulator can play, at least in the early stages, to propel the country forward at a faster pace in this endeavor. Ultimately, the objective should be to protect businesses from predatory practices to achieve a greater economic outcome.

## 7. Limitations and Future Studies

The fast advancement of technological innovation has now brought about the emergence of the adoption of cashless payment systems. This study investigates the issue of cashless payment systems, albeit with a set of limitations. First, this study was conducted from the perspective of businesses in Malaysia. Along these lines, the findings may not explicitly describe the adoption of cashless payment systems in different countries. Future examinations can evaluate cross-country to broaden their scope by using information from different countries. Secondly, the application of this study only reflects the target respondents' view at a particular time period. Accordingly, future academic scholars may consider using the longitudinal methodology, as it will give a scope for the analysis of span and evaluation of information. Thirdly, this study only selected businesses that have experience using cashless payment systems, and hence it could provide better knowledge of their adoption of cashless payment. Thus, a future study may select both experienced and inexperienced respondents to check whether there are any significant differences between them.

## 8. Conclusions

Malaysia's move towards a cashless society is a function of policy direction, as well as technology serving demand from certain market segments. The trend towards going cashless is a global one, and we should be aware that what works in one country may not apply to others due to differences in infrastructure (technology) and culture (behavior). The economic arguments are solid; for example, significant gains from cost and time savings will add percentages toward GDP growth, and cashless payments reduce opportunities for tax evasion, the shadow economy, and corruption. However, the size of the tax gap due to tax evasion and the shadow economy is usually hard to measure accurately. Less cash will also lead to less crime, but a cashless society will lead to increased cybercrime, electronic fraud, and digital crime/hacking. The use of encryption and strong authentication can help to reduce this.

Cashless payments will also lead to financial inclusion, especially for the non-banked population. The experience of developed countries shows that regulators play a role in advocating for financial inclusion. Accounts from other countries that have moved further along the cashless road note that there are certain segments of society that may now be excluded, such as the older generation that are not so tech-savvy (a generational gap) and communities with poor internet connectivity (the digital divide). In Africa, India, and China, technology is an enabler of financial inclusion. Reforms are happening out of economic necessity in developing countries.

**Author Contributions:** Conceptualization, M.R., I.I., S.B. and M.K.R.; methodology, M.R. and M.K.R.; software, M.K.R.; validation, M.R., M.K.R. and I.I.; formal analysis, M.K.R.; investigation, M.R., I.I. and S.B.; resources, I.I.; data curation, I.I., S.B. and M.R.; writing—original draft preparation, M.R., I.I., S.B. and M.K.R.; writing—review and editing, M.R. and I.I.; visualization, I.I.; supervision, M.R. and I.I.; project administration, M.R; funding acquisition, M.R. All authors have read and agreed to the published version of the manuscript.

**Funding:** This research received no external funding. The APC was funded by University of Sharjah.

**Institutional Review Board Statement:** Not applicable.

**Informed Consent Statement:** Informed consent was obtained from all subjects involved in the study.

**Data Availability Statement:** Data are private and confidential, as they concern a third party.

**Conflicts of Interest:** The authors declare no conflict of interest.

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
