# Peer review of "An Empirical Analysis of Cashless Payment Systems for Business Transactions"

_2199-8531, doi:10.3390/joitmc8040213_

Round 1

Reviewer 1 Report

The title of the paper is interesting. However, there is a lot of literature in the field.

The bibliography used in the literature review part is old. The authors must document themselves more on the new technologies.

The format of the paper makes reading difficult.

The conclusions part is poorly developed.

I recommend rewriting the paper and resubmitting it.

Author Response

Response to Reviewer 1 Comments

Point 1: The title of the paper is interesting. However, there is a lot of literature in the field.

The bibliography used in the literature review part is old. The authors must document themselves more on the new technologies.

Response 1: Thank you very much for your suggestions. We have improved the paper as per your suggestions.

Point 2: The format of the paper makes reading difficult.

Response 2: Thank you for the suggestion. We have tried to improve the structure of the paper.

Point 3: The conclusions part is poorly developed.

Response 3: Thank you. We have revised it.

Point 3: I recommend rewriting the paper and resubmitting it.

Response 2: Thank you. We have improved the entire paper.

Reviewer 2 Report

Dear authors, 

I believe this is an interesting piece of research. A few comments to make the paper even better: 

1) Please check this book: Building a Cashless Society The Swedish Route to the Future of Cash Payments https://library.oapen.org/handle/20.500.12657/23303 

I believe it would be beneficiary to your paper. 

2) A business model perspective could be covered. As a minor reflection since a lot has been written in this domain. Additionally, notion of an omni-channel strategy could be covered as well. How do mobile payments complement the existing type of payments? 

Ref:

Exploring the growth challenges of mobile payment platforms: A business model perspective https://www.sciencedirect.com/science/article/pii/S1567422319300857 

3) What are the drivers of such emerging technology (e.g., cashless payments) for industry-wide adoption (e.g., see ref about industry-wide adoption in the blockchain domain)? 

Ref:

Managing a blockchain-based platform ecosystem for industry-wide adoption: The case of TradeLens https://www.sciencedirect.com/science/article/pii/S0040162522005029

5) Improve Figure 1. Conceptual model.

I hope this helps. 

Author Response

Response to Reviewer 2 Comments

Point 1: Please check this book: Building a Cashless Society The Swedish Route to the Future of Cash Payments https://library.oapen.org/handle/20.500.12657/23303. I believe it would be beneficiary to your paper.

Response 1: Thank you very much for your valuable suggestions to improve our paper. We checked your reference book. Thank you.

Point 2: A business model perspective could be covered. As a minor reflection since a lot has been written in this domain. Additionally, notion of an omni-channel strategy could be covered as well. How do mobile payments complement the existing type of payments?

Ref: Exploring the growth challenges of mobile payment platforms: A business model perspective https://www.sciencedirect.com/science/article/pii/S1567422319300857

Response 2: Thank you for the suggestions. However, we believe that omni-channel strategy is out of the scope of this study, and future research can cover it.

We did cover the payments using mobile phones in this study (Please see page 8 and discussion section). Thank you.

Point 3: What are the drivers of such emerging technology (e.g., cashless payments) for industry-wide adoption (e.g., see ref about industry-wide adoption in the blockchain domain)?

Ref: Managing a blockchain-based platform ecosystem for industry-wide adoption: The case of TradeLens https://www.sciencedirect.com/science/article/pii/S0040162522005029

Response 2: Thank you very much for your suggestions. We have added some new literature.

Point 3: Improve Figure 1. Conceptual model. I hope this helps.

Response 2: Thank you very much for your suggestions. We have improved figure 1.

Reviewer 3 Report

1. The words in Figure1 should be adjusted to appropriate size.

2. Moderating variables and hypothesis were not described clearly.

(e.g. What types of industry? How many kinds of location or revenues the authors concern under this survey?

3. Yes, table 6 show the results of all the moderating effects, but the discussion and contribution base on the findings were not emphasized.

Author Response

Response to Reviewer 3 Comments

Point 1:  The words in Figure1 should be adjusted to appropriate size.

Response 1: Thank you very much for your suggestions. We have improved Figure 1.

Point 2: Moderating variables and hypothesis were not described clearly. (e.g. What types of industry? How many kinds of location or revenues the authors concern under this survey?

Response 2: Thank you very much for the comments. We have mentioned them under data collection and sample (please see page 6).

Point 3: Yes, table 6 show the results of all the moderating effects, but the discussion and contribution base on the findings were not emphasized.

Response 2: Thank you. The main objective of this study is to examine the antecedents of cashless payment systems among businesses in Malaysia; hence, the discussion on moderating variables is less emphasised (Please see page 14). We did revise it.

Round 2

Reviewer 1 Report

I think the paper has been improved. I think it can be published in this form.

Author Response

Thank you very much for your suggestions to improve the paper.